# The PROGRAM study: awake mapping versus asleep mapping versus no mapping for high-grade glioma resections: study protocol for an international multicenter prospective three-arm cohort study

Jasper Kees Wim Gerritsen  [ID] ,[1] Djaina D Satoer,[1]
Clemens Maria Franciscus Dirven,[1] Steven De Vleeschouwer,[2] Kathleen Seidel,[3]
Philippe Schucht,[3] Christine Jungk,[4] Sandro M Krieg,[5] Brian Vala Nahed,[6]
Mitchel Stuart Berger,[7] Marike Lianne Daphne Broekman,[8]
Arnaud Jean Pierre Edouard Vincent[1]

For numbered affiliations see end of article.

**Correspondence to**
Dr Jasper Kees Wim Gerritsen;
j.gerritsen@erasmusmc.nl

## ABSTRACT

**Introduction** The main surgical dilemma during glioma resections is the surgeon's inability to accurately identify eloquent areas when the patient is under general anaesthesia without mapping techniques. Intraoperative stimulation mapping (ISM) techniques can be used to maximise extent of resection in eloquent areas yet simultaneously minimise the risk of postoperative neurological deficits. ISM has been widely implemented for low-grade glioma resections backed with ample scientific evidence, but this is not yet the case for high-grade glioma (HGG) resections. Therefore, ISM could thus be of important value in HGG surgery to improve both surgical and clinical outcomes.

**Methods and analysis** This study is an international, multicenter, prospective three-arm cohort study of observational nature. Consecutive HGG patients will be operated with awake mapping, asleep mapping or no mapping with a 1:1:1 ratio. Primary endpoints are: (1) proportion of patients with National Institute of Health Stroke Scale deterioration at 6 weeks, 3 months and 6 months after surgery and (2) residual tumour volume of the contrast-enhancing and non-contrast-enhancing part as assessed by a neuroradiologist on postoperative contrast MRI scans. Secondary endpoints are: (1) overall survival and (2) progression-free survival at 12 months after surgery; (3) oncofunctional outcome and (4) frequency and severity of serious adverse events in each arm. Total duration of the study is 5 years. Patient inclusion is 4 years, follow-up is 1 year.

**Ethics and dissemination** The study has been approved by the Medical Ethics Committee (METC Zuid-West Holland/Erasmus Medical Center; MEC-2020–0812). The results will be published in peer-reviewed academic journals and disseminated to patient organisations and media.

## Strengths and limitations of this study

⇒ First multicentre prospective study directly comparing awake mapping, asleep mapping and no mapping for glioblastoma resections in or near eloquent areas.
⇒ International, multicenter design on a large scale, which will be of substantial benefit with regard to subgroup analyses and external generalisability of the results.
⇒ Observational design will not exclude all possible, inherent forms of bias.

**Trial registration number** ClinicalTrials.gov ID number NCT04708171 (PROGRAM-study), NCT03861299 (SAFE-trial)

## INTRODUCTION

Gliomas are the most common malignant tumours of the central nervous system and are classified into grades 1–4, where grades 1 and 2 consist of low-grade gliomas (LGG) and grades 3 and 4 represent high-grade gliomas (HGG).[1 2] Gliomas are relatively rare (incidence of 5/100 000 persons/year in Europe and North America), but are associated with a relatively high morbidity and mortality regardless of years of scientific efforts to improve clinical outcomes in these patients.[1–7]

Studies show that maximising the extent of resection of the contrast-enhancing part—and recently, the non-contrast-enhancing part as well—results in improved patient survival rates.[8–15] Moreover, patients with gross-total

resections derived the most benefit from the adjuvant chemoradiotherapy compared with patient with subtotal resections.[16] However, in excess of 50% of gliomas are located in or near eloquent areas of the brain.[2] Eloquent areas are important areas within the brain where speech and/or motor functions are located. Damaging these areas during surgery can lead to severe and permanent neurological deficits that seriously impact the quality of life. As a consequence of this worsened condition, some patients are excluded for radiotherapy and chemotherapy, leading to suboptimal clinical outcomes.[16]

Thus, the main surgical problem for the surgeon is the inability to accurately identify these eloquent areas when the patient is under general anaesthesia (GA) when no brain mapping techniques are being used. Surgeons often choose a more defensive approach for tumours that are located in or near these areas to prevent postoperative neurological deficits in patients with an already poor prognosis.[2 10 12–15] The use of intraoperative stimulation (neurophysiological) mapping techniques (ISM) can be necessary to enable the surgeon to resect as much tumour as possible while preserving quality of life and neurological functioning in these patients.[17] Mapping of motor-eloquent tumours can be performed while the patient is awake or asleep, while speech mapping can only be performed when the patient is awake. The use of mapping techniques has tremendous potential in glioma resections in eloquent areas, especially for HGG patients. However, there is currently no international consensus regarding the use of these techniques. The scientific evidence for the use of these techniques in this patient group is currently both inconclusive and fragmented. We therefore propose an international, multicenter prospective cohort study in which the use of awake and asleep mapping techniques in HGG patients will be evaluated.

The described research initiative will be able to study these techniques in a prospective setting while covering a breadth of centres and countries. Hence, the data generated in this ENCRAM (The European and North American Consortium and Registry for Intraoperative Stimulation Mapping) research collaboration will be able to answer multiple research questions with excellent generalisability, external validity and overall quality in both a cost-effective and practical setting.[18]

## METHODS AND ANALYSIS
### Study design
This is a international, multicenter, prospective, three-arm cohort study. Eligible patients are operated using awake mapping, asleep mapping or no mapping with a 1:1:1 ratio with a sequential computer-generated random number as subject ID. Patients with motor-eloquent tumours will be treated in all study arms, while speech-eloquent tumours will only be treated in either the awake mapping or no mapping arm. The PROGRAM study (prospective cohort study of high-grade glioma resections using awake craniotomy and intraoperative stimulation mapping) is similar to the SAFE-trial (SAFE surgery for glioblastoma patients: awake craniotomy versus craniotomy under general anesthesia: a multicenter randomized controlled trial) and is initiated by the same center, however, the presented study will be different in various ways: the PROGRAM study (1) will be an observational, prospective cohort study, (2) will include asleep mapping as an additional treatment arm, (3) will evaluate the extent of resection of the non-contrast-enhancing part of the tumour as well, (4) will include both WHO grade III and grade IV gliomas, (5) will include an oncofunctional score as one of the outcomes and (6) will include neurosurgical centers in the USA and is part of the ENCRAM Research Consortium.[18]

### Study objectives
The primary study objective is to evaluate the safety and efficacy of resections with or without mapping techniques (neurological morbidity and extent of resection) in HGG patients as expressed by National Institute of Health Stroke Scale (NIHSS) scores and volumetric data. Secondary study objectives are to study the overall survival (OS), progression-free survival (PFS) and oncofunctional outcome after resections with or without mapping techniques as expressed by survival data, progression on MRI scans and combining postoperative volumetric/functional outcomes, respectively.

### Study setting and participants
Patients will be recruited for the study from the neurosurgical or neurological outpatient clinic or through referral from general hospitals of the participating neurosurgical hospitals, located in Europe and the USA. The study is open to additional participating neurosurgical centres.

### Patient and public involvement statement
Patients enrolled in the SAFE-trial (awake craniotomy vs craniotomy under general anesthesia for glioblastoma patients, NCT03861299) were consulted for this study to include patient experiences with resections with and without mapping.

### Inclusion criteria
In order to be eligible to participate in this study, a subject must meet all of the following criteria:
1. Age ≥18 years and ≤90 years.
2. Tumour diagnosed as HGG (WHO grade III/IV) on MRI as assessed by the neurosurgeon.
3. Tumours situated in or near eloquent areas; motor cortex, sensory cortex, subcortical pyramidal tract, speech areas or visual areas as indicated on MRI (Sawaya grading II and II).[19]
4. The tumour is suitable for resection (according to neurosurgeon).
5. Written informed consent.

### Exclusion criteria
A potential subject who meets any of the following criteria will be excluded from participation in this study:
1. Tumours of the cerebellum, brainstem or midline.

2. Multifocal contrast enhancing lesions.
3. Medical reasons precluding MRI (eg, pacemaker).
4. Inability to give written informed consent.
5. Secondary HGG due to malignant transformation from LGG.
6. Second primary malignancy within the past 5 years with the exception of adequately treated in situ carcinoma of any organ or basal cell carcinoma of the skin.

## Interventions

(1) Awake craniotomy with local anaesthesia (arm 1: awake mapping).

On the evening before surgery 1.5–2.0 mg lorazepam is administered for anxiolysis and 2×8 mg dexamethason. The patient is sedated with a bolus injection of propofol (0.5–1 mg/kg) and kept sedated with a propofol infusion pump (mean: 4 mg/kg/hour) and remifentanil ((0.5–2 µg/kg/min). Supplemental $O_2$ might be provided through a nasal cannula. Patients typically receive 1–2 g of cefazolin and sometimes up to 1 g/kg of mannitol (all verified with the surgeon). The room is kept warm and patient covered as the goal is to have the core temperature above 36°C during motor mapping. An arterial line (with standard monitoring for vital signs in addition to BP monitoring), central venous catheter, and urinary catheter are inserted. The patient is awakened and positioned on the table. At this point local anaesthesia for the fixation of the head in the Mayfield clamp and the surgical field is provided with a mixture of 10 mL lidocaine 2% with 10 mL bupivacaine 0.5% plus epinephrine 1:200 000 for the Mayfield clamp and up to 40 mL bupivacaine 0.375% with epinephrine 1:200 000 for the surgical field. After positioning, clamp fixation and surgical field infiltration, patients are sedated again for the trephination until the dura mater is opened, after local application of some drops of local anaesthetics. Propofol sedation is stopped after opening of the dura, with the patient awakening with as few external stimuli as possible. Cortical stimulation is performed with a bipolar electrical stimulator. The distance between both poles is 5 mm, and stimulation is performed by placing this bipolar pincet directly on the cortical surface and stimulating with increasing electrical biphasic currents of 2–12 mA (1–2 mA increasing steps, pulse frequency 60 Hz, single pulse phase duration of 100 µs) until motor or speech arrest is observed. For motor mapping a 2 s train and for speech mapping a 5 s train is used, respectively. The intraoperative neurolinguistic test-battery is performed by a neuropsychologist/linguist, who will inform the neurosurgeon of any kind of speech arrest or dysarthria. The difference between these is not always clear, but can be distinguished from involuntary muscle contraction affecting speech. When localising the motor and sensory cortex, the patient is asked to report any unintended movement or sensation in extremities or face. Confirmed functional cortical areas are marked with a number. After completion of cortical mapping, a resection of the tumour is performed as radical as possible using an ultrasonic aspirator and suction tube, while sparing these functional areas. When the tumour margins or white matter is encountered or when on regular neuronavigation the eloquent white matter tracts are thought to be in close proximity, subcortical stimulation (biphasic currents of 8–16 mA, 1–2 mA increasing steps, pulse frequency 60 Hz, single pulse phase duration of 100 µs, 2 s train) is performed to localise functional tracts. If subcortical tracts are identified, resection is stopped. During the resection of the lesion close to an eloquent area, the patient is involved in a continuous dialogue with the neuropsychologist. That way the neurosurgeon has 'online'-control of these eloquent areas. In case of beginning disturbances of communication or of motor or sensory sensations the resection is cessated immediately. When, due to stimulation, an epileptic seizure occurs, this is stopped by administering some drops of iced saline on the just stimulated cortical area. If a seizure continues, an intravenous propofol or diphantoin bolus of 0.5 mg/kg is administrated and repeated until the seizure stops. The mapping procedure is temporarily halted. If the patient is adequate, cooperative and able to carry out tasks after the seizure, the mapping procedure can continue. In the case of refractory seizures, the mapping procedure will be permanently halted and the resection will continue under GA. After resection of the tumour a final neurological examination is performed. During closure of the surgical field the patient is sedated with propofol again. After wound closure and dressing, sedation is stopped. The awake patient is transferred to the post-anaesthesia care unit (PACU), where the patient is haemodynamically and neurologically monitored for 24 hours.

(2) Asleep mapping under GA (arm 2: asleep mapping).

UCSF protocol: an intravenous is started on ipsilateral hand to the tumour. The patient is premedicated with up to 2 mg of midazolam. None if altered mental status (prevent further increase in ICP). Arterial (ipsilateral to tumour) catheter is inserted after induction of anaesthesia. Anaesthesia goals are to decrease ICP (if high), to maintain adequate CPP (at least 70 mm Hg) to prevent cerebral ischaemia from brain retraction, and to allow intraoperative cortical motor mapping. Patients typically receive 1–2 g of cefazolin, and 4 mg of decadron before skin incision, and sometimes up to 1 g/kg of mannitol (all verified with the surgeon). The room is kept warm and patient covered as the goal is to have the core temperature above 36°C during motor mapping. Induction with propofol. In case of increased ICP, have patient hyperventilate during preoxygenation and continue hyperventilation with mask as soon as possible after induction of anaesthesia. Fentanyl up to 5 µg/kg in divided doses throughout induction, prior to intubation. Adequate neuromuscular blockade (rocuronium) is verified prior to intubation to avoid coughing/straining. Eyes are taped, and at least one additional large bore intravenous is inserted. Neuromuscular relaxation is let to wear off for motor mapping (do not reverse). The patient position will depend on location of tumour. Anaesthesia is mainained with 70% nitrous oxide in oxygen, low-dose

inhalation agent (less than 0.5 MAC), and a remifentanil (0.2 µg/kg/min) or fentanyl infusion (2 µg/kg/hour). Euvolaemia is maintained (Lactated Ringer's). Mild hyperventilation (PaCO2 35 mm Hg) is used. Once the bone flap is removed, the surgeon assesses the tightness of the dura. ICP is further decreased if necessary (pCO2, mannitol, propofol, head up, etc). Once the dura is open, the goal is to avoid brain shift so that stereotactic navigation system can be used optimally. During motor mapping, the arm, leg and face are uncovered to observe for movement. Stimulation is performed with the use of evoked potentials and continuous dynamic mapping/direct subcortical stimulation (CDM) with a monopolar stimulator (INOMED Medizintechnik, Germany). During stimulation, TES/MEP (transcranial evoked stimulation/motor evoked stimulation) registration is performed of the contralateral m. orbicularis oris, m. orbicularis oculi, m. biceps brachii, m. abductor pollicis, m. rectus femoris and m. tibialis anterior; and the ipsilateral m. abductor pollicis. SSEP (somatosensory evoked stimulation) registration is performed of the contralateral n. tibialis and bilateral n. medianus. The pulse form is negative, with five pulses and a pulse width of 500 µs, ISI 4 and current between 5 and 20 mA. In case of poststimulation continuation of motor activity, the surgeon will try to stop it by applying cold saline on the cortex. Have propofol (10 mg/mL) in line in case of intraoperative seizures (0.5 mg/kg for seizure suppression). May use neuromuscular relaxants after the last motor mapping. Fentanyl infusion is usually stopped at the beginning of closure. Remifentanil infusion is stopped about 10 min before end of surgery. At this point, use of inhalation agent may be replaced with a propofol infusion (50–100 µg/kg/min). pCO2 is normalised to facilitate spontaneous breathing at the end of the operation. Use of inhalation agents (or propofol) is usually stopped about 10–15 min before end of surgery, and nitrous oxide at the end of surgery. Residual neuromuscular blockade is reversed once the Mayfield pins have been removed. At the end of the procedure all anaesthetics are stopped and patient is brought to the PACU (PACU/IC). Detubation of the patient is performed as early as possible, if patient fulfils the detubation criteria (>36°C body temperature, stable haemodynamics, sufficient spontaneous ventilation, adequate response to verbal orders). Postoperative analgesia is provided with paracetamol intravenous or p.o. 1 g up to 4 dd and morphine 7.5 mg SC up to 4 dd, if necessary. At the PACU the patient is haemodynamically and neurologically monitored for 24 hours.

Bern protocol: the following details are different to the above-mentioned protocol. Total intravenous anesthesia without inhalation agents is used (TIVA-only). A bolus of propofol is started (1–2 mg/kg body weight) with fentanyl (1–2 mg/kg body weight), and remifentanil (1–2 mg/kg body weight) and maintained with propofol (100–200 mg/kg/min) and remifentanil (0.5 mg/kg/h). A short-acting relaxant is used (Esmeron 0.6 mg/kg body weight for the purpose of intubation). Then, the 'train-of-four'

technique is used involving percutaneous stimulation of the right median nerve (40 mA, 0.2- msec pulse duration) to test recovery from muscle relaxation. MEPs are recorded from subdermal electrodes in order to quantify the evoked responses. A combination of DCS MEP via a four-contact strip electrode placed on the pre-central gyrus for focal and selective stimulation and a back-up TES MEP via scalp electrodes is used.[20] The 'suction probe' (INOMED medizintechnik, Germany; #525 650) is used for cortical mapping and subcortical continuous dynamic mapping.[21] For subcortical stimulation a monopolar cathodal pulse stimulation is used with train of 5 pulses of 0.5 msec duration, ISI 4 msec and 2 Hz repetition rate. The mapping intensities range from 20 mA down to 3 mA (and in selective cases down to 1mA). Monitoring motor function is continued until dura closure in order to detect vascular injuries (for instance due to vasospasms).

(3) Craniotomy under GA without mapping (arm 3: no mapping).

On the evening before surgery, 1.5–2.0 mg lorazepam is administered for anxiolysis.Sixtey min before anaesthesia induction, the patient receives 1 g paracetamol orally and 7.5–15 mg midazolam orally if requested for sedation. En route to the operating room, 0.5–2 mg midazolam intravenous may be given. 1 g cefazoline is given intravenous for antibiotic prophylaxis before anaesthesia induction. GA is induced intravenously with fentanyl 0.25–0.5 mg, propofol 100–200 mg and cis-atracurium 10–20 mg. After induction of anaesthesia, the patient is orotracheally intubated and mechanical ventilation is applied. Respiratory rate and tidal volume are adjusted to keep the patient normocapnic.

An arterial line (alternatively: two peripheral intravenous), central venous catheter (v. basilica) and urinary catheter are inserted. Anaesthesia is maintained with propofol (up to 10 mg/kg/hour) and remifentanil (0.5–2 µg/kg/min). isoflurane (up to 1 MAC) and clonidine (1–2 µg/kg) may be added for maintenance, if necessary (a beta blocker or calcium channel blocker may be used to control BP as an alternative to clonidine). The fluid management is aiming for normovolaemia. 0.9% saline solution and balanced crystalloids are used for maintenance, in case of blood loss >300 mL, HAES 130/0.4 solution (hydroxyethyl starch) will be given. Temperature management is aiming for normothermia, warm-air blankets and warmed infusion lines are used. Arterial blood gas analysis is performed at the beginning of the procedure and repeated, if necessary. Electrolytes are controlled and substituted and hyperglycaemic will be treated with insulin, if necessary. The anaesthetised patient is positioned on the table. Local infiltration of the scalp is performed with 20 mL lidocaine 1% with epinephrine 1:200.000 to reduce bleeding. The insertion points of the Mayfield clamp are not infiltrated with local anaesthetics.

Trephination and tumour resection are performed without any additional neuro-psychological monitoring, guided by standard neuronavigation. At the end of the

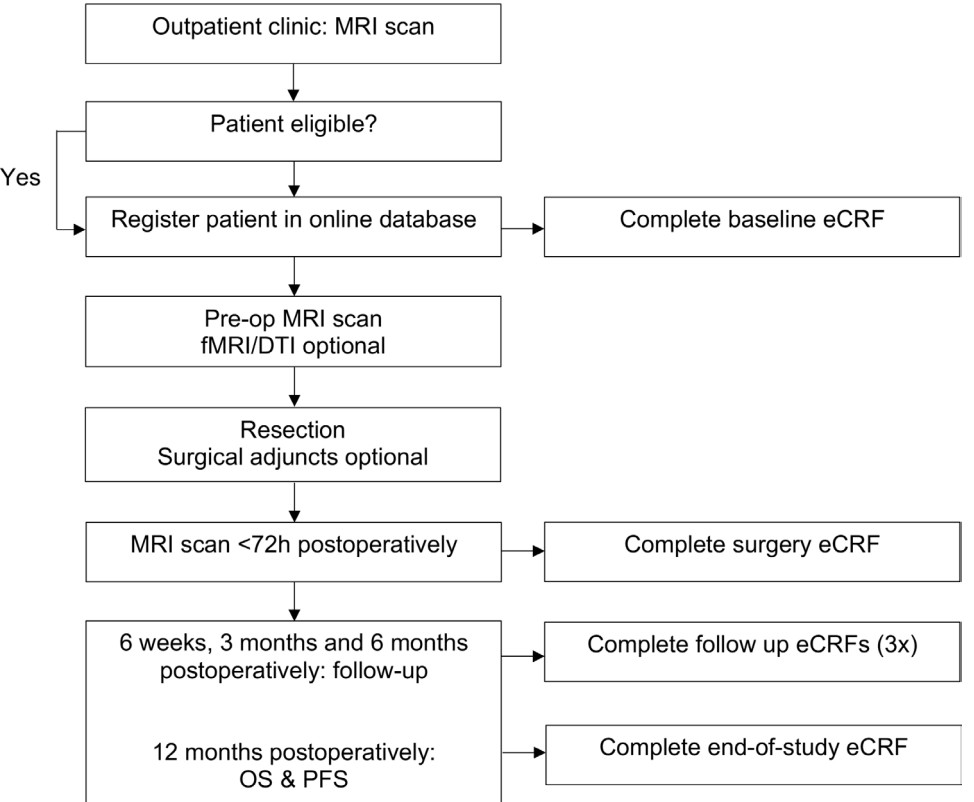

**Figure 1** Study flow diagram. DTI, diffusion tensor imaging; OS, overall survival; PFS, progression-free survival.

procedure, all anaesthetics are stopped and patient is brought to the PACU. Detubation of the patient is performed as early as possible, if patient fulfils the detubation criteria (>36°C body temperature, stable haemodynamics, sufficient spontaneous ventilation, adequate response to verbal orders). Postoperative analgesia is provided with paracetamol intravenously or orally 1 g up to 4 dd and morphine 7.5 mg SC up to 4 dd, if necessary. At the PACU the patient is haemodynamically and neurologically monitored for 24 hours.

### Surgical adjuncts and additional imaging
The use of fMRI, Diffusion Tensor Imaging (DTI), ultrasound or 5-ALA (5-aminolevulinic acid) is allowed to be used in all groups on the surgeon's indication.

### Participant timeline
The flow diagram illustrates the main study procedures, including follow-up evaluations (figure 1). In summary, study patients are allocated to either the awake mapping, asleep mapping or no mapping group and will undergo evaluation at presentation (baseline) and during the follow-up period at 6 weeks, 3 months, 6 months and 12 months postoperatively. Motor function will be evaluated using the NIHSS and Medical Research Council (MRC) scales. Language function will be evaluated using a standard neurolinguistic test-battery consisting of the Aphasia Bedside Check (ABC), Shortened Token Test, Verbal fluency, Picture description and Object naming. Cognitive function will be assessed using the Montreal

Cognitive Assessment (MoCA). Patient functioning with be assessed with the Karnofsky Performance Scale (KPS) and the ASA (American Society of Anesthesiologists) physical status classification system. Health-related quality of life (HRQoL) will be assessed with the EQ-5D questionnaire (EuroQol Five Dimensions health questionnaire) and the QLQ-C30 and QLQ-BN20 questionnaires. OS and PFS will be assessed at 12 months postoperatively. We expect to complete patient inclusion in 4 years. The estimated duration of the study (including follow-up) will be 5 years.

### Study procedures: clinical evaluations and follow-up
- ► Preoperative (baseline) CRF (Case Report Form).
  - – Unique subject ID, demographics (centre, year, gender, age), tumour-specific factors (tumour volume preoperative, tumour hemisphere and lobe; eloquent areas), patient specific factors: preoperative KPS, ASA score, neurological status (NIHSS), MRC grade arm/leg (for motor-eloquent tumors), neurolinguistic testing, MOCA, QoL questionnaires (QLQ-C30, QLQ-BN20, EQ-5D).
- ► Surgery CRF
  - – Type of ISM, surgeon's rationale for modality, surgeon's goal, use of preoperative steroids (if yes: clinical improvement, conversion to mapping possible); use of surgical adjuncts (if DTI: integrity of tracts), use of additional imaging, radiological factors: resection percentage (both the contrast-enhancing

and non-contrast-enhancing part), residual volume and postoperative ischaemia.

► Follow-up CRFs
- Six weeks postoperatively: histology and molecular markers (WHO grade, MGMT status, IDH-1 status), neurological status (NIHSS), MRC grade arm/leg, status MRC arm/MRC leg/facialis/speech/visual (new, worsened, improved, stable), KPS, QoL questionnaires (QLQ-C30, QLQ-BN20, EQ-5D).
- Three months postoperatively: neurological status (NIHSS), MRC grade arm/leg, status MRC arm/MRC leg/facialis/speech/visual (new, worsened, improved, stable), KPS, neurolinguistic testing, MOCA, QoL questionnaires (QLQ-C30, QLQ-BN20, EQ-5D).
- Six months postoperatively: neurological status (NIHSS), adjuvant treatment, MRC grade arm/leg, status MRC arm/MRC leg/facialis/speech/visual (new, worsened, improved, stable), KPS, MOCA, QoL questionnaires (QLQ-C30, QLQ-BN20, EQ-5D).
- Twelve months postoperatively: PFS, OS (end of study).

## Outcomes
### Primary outcome measures
The primary outcomes are (1) proportion of patients with NIHSS deterioration at 6 weeks, 3 months and 6 months postoperatively; deterioration is defined as an increase of at least one point on the total NIHSS score compared with this score at baseline and (2) residual tumour volume of the contrast-enhancing and non-contrast enhancing part, as assessed by a neuroradiologist on postoperative T1 with contrast and T2/FLAIR MRI scan sequences using manual or semiautomatic volumetric analyses (Brainlab Elements iPlan CMF Segmentation, Brainlab AG, Munich, Germany or similar software).

### Secondary outcome measures
The secondary outcomes are (1) PFS at 12 months defined as time from diagnosis to disease progression (occurrence of a new tumour lesions with a volume greater than 0.175 $cm^3$, or an increase in residual tumour volume of more than 25%) or death, whichever comes first; (2) OS at 12 months defined as time from diagnosis to death from any cause; (3) oncofunctional outcome defined as the calculated coordinate of the extent of resection (or residual volume) on the x-axis and the postoperative NIHSS deterioration (delta NIHSS) on the y-axis and (4) frequency and severity of serious adverse events (SAEs) in each arm.

## National Institute of Health Stroke Scale
The NIHSS, or NIH Stroke Scale is a tool used by healthcare providers to objectively quantify the impairment caused by a stroke, but has been used extensively for outcome in glioma surgery because of the lack of such scale for neuro-oncological purposes and has been validated. The NIHSS is composed of 11 items, each of which

scores a specific ability between a 0 and 4. For each item, a score of 0 typically indicates normal function in that specific ability, while a higher score is indicative of some level of impairment. The individual scores from each item are summed in order to calculate a patient's total NIHSS score. The maximum possible score is 42 and the minimum score 0.

## Aphasia bedside check
ABC is a short screening test to detect aphasic disturbances at language comprehension and language production level at the main linguistic levels. It consists of 14 items in total. The cut-off score for signs of aphasia is ≤12.

## Shortened Token Test
The shortened Token Test is a test for language comprehension and for the severity of a language disorder. The patient is asked to point and to manipulate geometric forms on verbal commands. It consists of 36 items. The cut-off score is 29.5.

## Verbal fluency (category and letter)
Category and letter fluency are tests to assess flexibility of verbal semantic and phonological thought processing, semantic memory and concept generation. The patients is asked to produce words of a given category (animals, professions) or beginning with a given letter (D, A, T) within a limited time span.

## Picture description and object naming
The picture description test, a subtest from the CAT-NL (Comprehensive Aphasia Test-Netherlands), is used to assess semispontaneous speech in an oral and written way (5 min each condition). Scoring can be done according to the manual or more thoroughly according to the variables mentioned by Vandenborre et al.[22] To assess word retrieval, various object naming tests are used: BNT (Boston Naming Test), DuLIP (Dutch Linguistic Intraoperative Protocol) and VAN-POP (Verb and Noun test for Perioperative Testing).

## Montreal Cognitive Assessment
The MOCA is a cognitive screening test to detect mild impairments across several cognitive domains; attention, verbal memory, language, visuoconstructive skills, conceptual thought, calculation and orientation. The total score is 30, the cut-off score is ≤26.

## EQ-5D
The EQ-5D is a standardised questionnaire to assess the general HRQoL in five domains: mobility, self-care, usual activity, pain/discomfort and anxiety/depression. It is developed by the EuroQol Group and can also be used to calculate quality-adjusted life years for cost–utility analyses.

## QLQ-C30 and QLQ-BN20

The QLQ-C30 and QLQ-BN20 are standardised questionnaires that have been designed by the Euoprean Organisation for Research and Treatment of Cancer (EORTC). They are used to assess the quality of life in cancer patients in general (C30) and brain tumor patients (BN20) by incorporating functional scales (physical, role, cognitive, emotional, social) and symptom scales (fatigue, pain, nausea and vomiting, seizures, communicating).

## Sample size

This study has two primary endpoints. In order to guarantee that the overall type I error rate does not exceed 5%, we apply a weighted Bonferroni correction for multiple testing. The sample size calculations that follow take that into account. For the first primary endpoint, proportion of patients with neurological deterioration at 6 weeks postsurgery, we assume a deterioration rate of 10% in the control group (arm 3: no mapping), and 3% in the experimental groups (arm 1 and 2: awake and asleep mapping). A two-sample test for proportions with continuity correction requires 411 patients (137 per arm) in total in order to detect the above-mentioned difference of 7% with 80% power at a 4% significance level. For the second primary endpoint, proportion of patients without residual contrast-enhancing tumour on postoperative MRI, we assume a success rate of 25% in the control group (arm 3: no mapping), and 50% in the experimental groups (arm 1 and 2: awake and asleep mapping). A two-sample test for proportions with continuity correction requires 188 patients (94 per arm) in total in order to detect the above-mentioned difference of 25% with 80% power at a 1% significance level. In order to power the study for both primary endpoints, we should include the larger required number of patients, that is, 411. A total of 411 eligible and evaluable patients in three arms allow the difference of 25% in proportion of patients without residual tumour to be detected with 88% power. Taking into account possible ineligibility and withdrawal of consent (we estimate this at 10%), a total of 453 patients will be included (151 patients per arm).

## Data collection

All patient data are collected using the electronic data software Castor EDC (Electronic Data Capture). This software allows built-in logical checks and validations to promote data quality. Data entry and group allocation is performed by the study coordinator or locally by trained physicians and research nurses under supervision of the local investigator.

## Data analysis

All analyses will be according the intention to treat principle, restricted to eligible patients. Patients initially registered but considered ineligible afterwards based on the histological analysis on tissue extracted during surgery, will be excluded from all analyses.

## Primary study parameters

The primary endpoints will be analysed using multivariate logistic regression. Subgroup analyses for tumour grade (WHO grade III/IV), preoperative neurological morbidity, preoperative tumour volume, patient's age (in 10-year age brackets) and tumour location/eloquence will be performed.

We will be including a stratification factor in the primary analysis model with each 10 observed events using the order of prognostic value as mentioned in the paragraph above, where the first 10 events will be used to estimate the effect of the arm. This rule will be applied in case less than 40 patients in total develop neurological deterioration. In the so constructed multivariate logistic regression model the treatment arm effect will be tested at 4% significance level. The primary analysis of proportion of patients without residual contrast-enhancing and non-contrast-enhancing tumour consist of a multivariate logistic regression, where arm effect is corrected for all minimisation factors. In this model the group effect will be tested at 1% significance level. Manual or semiautomatic segmentation will be performed on axial T1 and T2/FLAIR MRI contrast enhanced slices to measure preoperative and postoperative tumour volume. A determination of volumes will be calculated blinded for the treatment group.

## Secondary study parameters

The Kaplan-Meier method will be used to estimate PFS and OS proportions per treatment group at appropriate time points, while the Greenwood estimate of the SE (standard error) will be used to construct the corresponding 95% CI. Multivariate cox proportional hazards models will be built for PFS and OS where treatment group effect will be corrected for minimisation factors age group (≤55 years vs >55 years), KPS (80–90 vs >90) and left or right hemisphere. Additionally, competing risk analysis will be used to calculate cumulative incidence of PFS (with competing risks progression/relapse and death without progression/relapse which add up to 100% at every time point). Oncofunctional outcome will be evaluated using a scatter or bubble plot with volumetric data on the x-axis and neurological status (NIHSS) or patient performance (KPS) on the y-axis. SAEs in both groups will be described.

## Study monitoring

No scheduled on-site monitoring visits will be performed. Local investigators will remain responsible for the fact that the rights and well-being of patients are protected, the reported trial data are accurate, complete and verifiable from source documents and the conduct of the trial is in compliance with the currently approved protocol/amendment(s), with GCP (Good Clinical Practice) and with the applicable regulatory requirement(s). Direct access to source documentation (medical records) must be allowed for the purpose of verifying that the data recorded in the CRF are consistent with the original source data. No data safety monitoring board will be

installed: all interventions are care-as-usual and patients are allocated without randomisation.

## Adverse events and SAEs

AEs (adverse events) are defined as any undesirable experience occurring to a subject during the study, whether or not considered related to neurosurgery. All AEs reported spontaneously by the subject or observed by the investigator or his staff will be recorded from start of surgery until 6 weeks after surgery. SAEs (serious adverse events) are any untoward medical occurrence or effect that results in death; is life-threatening (at the time of the event); requires hospitalisation or prolongation of existing inpatients' hospitalisation; results in persistent or significant disability or incapacity or any other important medical event that did not result in any of the outcomes listed above due to medical or surgical intervention, but could have been based on appropriate judgement by the investigator. An elective hospital admission will not be considered as an SAE. Most of the (serious) adverse effects of treatments be mainly related to the surgery: post operative pain, nausea and anaemia (in case of massive blood loss), Infections, intracranial haemorrhage, epilepsy, aphasia, paresis/paralysis in arms or/and legs.

Most of the (serious) adverse effects of treatments (awake surgery or surgery under generalised anaesthesia) will be mainly related to the surgery: postoperative pain, nausea and anaemia (in case of massive blood loss), infections, intracranial haemorrhage, epilepsy, aphasia, paresis/paralysis in arms or/and legs. The neurological morbidity is under investigation in this trial and well-known risk/complications of the craniotomy and can be attributed to the nature of the operation. Neurosurgical clinics are well adapted to prevent and treat such events. SAEs will be collected through routine data management.

## Publication of results

Trial results will be published in an international journal, communicated to neurological and neurosurgical associations and presented at (inter)national congresses.

## ETHICS AND DISSEMINATION

The study has been approved by the Medical Ethics Committee (METC Zuid-West Holland/Erasmus Medical Center; MEC-2020–0812) and is conducted in compliance with the European Union Clinical Trials Directive (2001/20/EC) and the principles of the Declaration of Helsinki (2013). The results of the study will be published in peer-reviewed academic journals and disseminated to patient organisations and media.

## DISCUSSION

Neurosurgeons face a major dilemma during glioma surgery: maximising extent of resection while minimising risk of postoperative neurological deficits. The use of awake or asleep mapping techniques has the potential to equip the surgeon intraoperatively with the needed information to balance these two surgical goals.

A substantial amount of evidence is available on the usefulness of awake mapping to increase resection percentage while preserving quality of life in LGG patients.[23–34] In contrast, only very few studies have reported the use of awake mapping in HGG patients, although this technique could be of important value in these patients as well.[17 23 25–27 34] Recent retrospective evidence showed that glioblastoma patients operated with awake mapping had significant less postoperative neurological morbidity and significantly higher percentage of total resections.[35 36] In patients with motor-eloquent tumours, the use of asleep mapping techniques with evoked potentials or CDM can be a viable alternative to preserve these functional tracts.[20 37 38]

There is a clear need for solid prospective evidence of the use of these techniques in HGG patients. The presented international neurosurgical research consortium will provide the needed infrastructure to perform ongoing large-scale data collection.[18] This study aims to evaluate whether the use of awake or asleep mapping is the appropriate answer to the surgeon's surgical dilemma during HGG resections. Furthermore, it will be the first to directly compare awake and asleep mapping techniques in their ability to improve patient outcomes for neurological morbidity, quality of life and survival. Last, using various multivariate analyses, there will be an additional focus on identifying the best surgical choice in subgroups of HGG patients.

## Trial status

The study will start at 1 April 2021 and is open to additional participating neurosurgical centers.

**Author affiliations**

[1]Department of Neurosurgery, Erasmus MC, Rotterdam, The Netherlands

[2]Department of Neurosurgery, Katholieke Universiteit Leuven/UZ Leuven, Leuven, Belgium

[3]Department of Neurosurgery, Inselspital Universitätsspital Bern, Bern, Switzerland

[4]Department of Neurosurgery, Universitätsklinikum Heidelberg, Heidelberg, Germany

[5]Department of Neurosurgery, Technical University of Munich, Munich, Bayern, Germany

[6]Department of Neurosurgery, Massachusetts General Hospital, Boston, Massachusetts, USA

[7]Department of Neurosurgery, University of California San Francisco, San Francisco, California, USA

[8]Department of Neurosurgery, Medisch Centrum Haaglanden, The Hague, The Netherlands

**Contributors** JKWG, AJPEV and MLDB designed the study, wrote the study protocol and are end-responsible for the implementation and organisation of the study in all participating centres. JKWG wrote the study protocol and is responsible for the implementation and organisation of the study in all participating centres and the conduct of the database. DDS wrote the neuro-linguistic protocol and is responsible for the implementation of this protocol in all participating centers. CMFD contributed to the design of the study. SDV contributed to the design of the study and is responsible for the local conduct of the study in Leuven. PS and KS contributed to the design of the study and are responsible for the local conduct of the study in Bern. CJ contributed to the design of the study and is responsible for the local conduct of the study in Heidelberg. SMK contributed to the design of the study and is responsible for the local conduct of the study in Munich. BVN

contributed to the design of the study and is responsible for the local conduct of the study in Boston. MB contributed to the design of the study and is responsible for the local conduct of the study in San Francisco. MLDB contributed to the design of the study and is responsible for the local conduct of the study in The Hague. All authors read and approved the final version of the manuscript.

**Funding** The authors have not declared a specific grant for this research from any funding agency in the public, commercial or not-for-profit sectors.

**Competing interests** None declared.

**Patient and public involvement** Patients and/or the public were involved in the design, or conduct, or reporting, or dissemination plans of this research. Refer to the Methods section for further details.

**Patient consent for publication** Not required.

**Provenance and peer review** Not commissioned; externally peer reviewed.

**ORCID iD**
Jasper Kees Wim Gerritsen http://orcid.org/0000-0003-4619-865X

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
