## [Reviewer comments · BMJ Open]

ARTICLE DETAILS

TITLE (PROVISIONAL)	The PROGRAM-study: Awake Mapping versus Asleep Mapping versus No Mapping for High-Grade Glioma Resections: Study Protocol for An International Multicenter Prospective 3-Arm Cohort Study
AUTHORS	Gerritsen, Jasper; Dirven, Clemens; De Vleeschouwer, Steven; Schucht, Philippe; Jungk, Christine; Krieg, Sandro; Nahed, Brian; Berger, Mitchel; Broekman, Marike; Vincent, Arnaud

VERSION 1 – REVIEW

REVIEWER	Ruiz-Garcia, Henry Mayo Clinic, Neurosurgery
REVIEW RETURNED	29-Dec-2020

GENERAL COMMENTS	This trial is a very interesting study with potential deep implications in glioma surgery. There are minor details I would appreciate could be addressed throughout the manuscript. Please define the group of patients will be included in the study, it is not clear if they will be only GBM patients or also grade III glioma patients. Consider modify the title if only patients with grade IV tumors will be evaluated. Study Objectives: I would consider rewording this section, the objectives of this study are not to improve safety, efficacy, or OS ad PFS. The trial is designed to study if any of the arms could offer any of these benefits. Please consider avoiding the use of the term "(sub)cortical", I would rather use "cortical and subcortical" Page 8, line 14: there is a typo in "brainstem" Participant timeline: Are the patients will be evaluated only at 12 months? or also at 6 months as stated in the abstract? Interventions: Please elaborate on the following points: Arm 1: awake mapping: Which is the plan to increase the stimuli (current will be increase every 0.5 mA? Or every 1 mA?, etc.)? Are you planning to confirm any positive finding? How are you going to do it? Could you please elaborate on the cases that will require ECoG? Which kind of grids will be used? Are the grids going to be the same for all the institutions? Will the mapping continue after refractory seizures requiring propofol? Which criteria will be used to decide it is safe/adequate to continue? Could you please
---

	elaborate on the OR temperature, the use of steroids and antibiotics as it was done for Arm 2? Outcomes: Primary Outcomes: I would suggest elaborating on the following: Which is the plan to perform the volumetrics? Which MRI sequence(s) will be used? Which software will be used? Are they going to be performed by each institution? Secondary Outcomes: Will OS and PFS be evaluated also at 6 months? (please review the abstract) Sample size: Could you please clearly define, and state in the manuscript, which groups will be the control and experimental group? Data Analysis: Secondary parameters: will you consider the Bonferroni correction for multiple comparison? (>2 groups).
--	--

REVIEWER REVIEW RETURNED	Flannery, Thomas Royal Victoria Hospital, Neurosurgery 12-Jan-2021
--	---

GENERAL COMMENTS	Thank you for the opportunity to review this study which will address an important question: is mapping (either asleep or awake) associated with better outcomes in high-grade glioma patients with eloquent tumours compared with standard asleep craniotomy. Reviewer comments for “The PROGRAM-study: Awake Mapping versus Asleep Mapping versus No Mapping for High-Grade Glioma Resections: Study Protocol for An International Multicenter Prospective 3-Arm Cohort Study”. The co-investigators of this study are highly renowned and very much respected in this field which gives the study tremendous strength, validity and international “robustness”. This will be an important study to address the issue of whether mapping techniques (either asleep or awake) leads to better patient outcomes (including improved EOR rates, lower disability and longer survival) compared with asleep craniotomy for eloquently located high-grade gliomas. I particularly liked the authors’ use of an onco-functional score that could have a lot of potential in helping predict a threshold of “maximal safe resection” with optimal survival and quality of life benefits. With all due respect to the authors, I hope they don’t mind me highlighting a few minor typographical errors and ask a few questions that I’m sure they can easily answer in terms of study design. The study is quite similar to the SAFE trial and in the technical details in this study two arms are mentioned instead of three. Could this be amended please in the methods and results
---

sections? Also I wondered if the statistical analysis needs to be reviewed by a statistician as the SAFE study has 2 study arms while the Program study has three.

There are a few other minor typo errors outlined in the methods and results section.

Abstract:

spelling mistake “mppaing”.

Change “Though” to “However, in excess of 50% of gliomas

Introduction: use “breadth” not “breath”

Methods:

nasal “cannula” not “canular”

Study procedure:

Could “Description of **presenting** symptoms....” at follow-up assessments be replaced with “Description of any current (new or ongoing) symptoms...?” (use of the word “Presenting” implies symptoms at the time of diagnosis).

In terms of questions about the study design I wondered about the following:

Primary outcome measures: will there be any analysis of non-enhancing tumour volume pre- and post-surgery?

How useful is asleep ISM for tumours near speech centres?

Will there be stratification or subgroup analysis based on tumour volume and WHO grade III/IV?

What if an awake craniotomy has to be converted to an asleep craniotomy? I presume this patient will then be included in the asleep group??

VERSION 1 – AUTHOR RESPONSE

Reviewer: 1

This trial is a very interesting study with potential deep implications in glioma surgery. There are minor details I would appreciate could be addressed throughout the manuscript.

Please define the group of patients will be included in the study, it is not clear if they will be only GBM patients or also grade III glioma patients. Consider modify the title if only patients with grade IV tumors will be evaluated.

Both grade III and IV will be included (high-grades). We have clarified this in the inclusion criteria

Study Objectives: I would consider rewording this section, the objectives of this study are not to improve safety, efficacy, or OS ad PFS. The trial is designed to study if any of the arms could offer any of these benefits.

Valuable comment. We have adjusted this section accordingly.

Please consider avoiding the use of the term "(sub)cortical", I would rather use "cortical and subcortical"
Adjusted accordingly

Page 8, line 14: there is a typo in "brainstem"
Adjusted.

Participant timeline: Are the patients will be evaluated only at 12 months? or also at 6 months as stated in the abstract?

Patients will be evaluated at 6 weeks, 3 months, 6 months and 12 months. We have clarified this schedule at the section 'study procedures: clinical evaluations and follow-up' in the text and as an illustration (flowchart).

Interventions: Please elaborate on the following points:

Arm 1: awake mapping: Which is the plan to increase the stimuli (current will be increase every 0.5 mA? Or every 1 mA?, etc.)? Are you planning to confirm any positive finding? How are you going to do it? Could you please elaborate on the cases that will require ECoG? Which kind of grids will be used? Are the grids going to be the same for all the institutions? Will the mapping continue after refractory seizures requiring propofol? Which criteria will be used to decide it is safe/adequate to continue? Could you please elaborate on the OR temperature, the use of steroids and antibiotics as it was done for Arm 2?

Thank you for your comment. We have elaborated on these topics in the intervention section.

Outcomes:

Primary Outcomes: I would suggest elaborating on the following: Which is the plan to perform the volumetrics? Which MRI sequence(s) will be used? Which software will be used? Are they going to be performed by each institution?

The volumetric analysis will be done using BrainLab iPlan software on T1+Gd MRI sequences. Added to the manuscript.

Secondary Outcomes: Will OS and PFS be evaluated also at 6 months? (please review the abstract)
OS/PFS will be evaluated at 12 months. Adjusted accordingly at both places.

Sample size: Could you please clearly define, and state in the manuscript, which groups will be the control and experimental group?

Adjusted accordingly.

Data Analysis:

Secondary parameters: will you consider the Bonferroni correction for multiple comparison? (>2 groups).
Yes, the Bonferroni correction will be used for both the primary and secondary outcome analyses.

Reviewer: 2

Thank you for the opportunity to review this study which will address an important question: is mapping (either asleep or awake) associated with better outcomes in high-grade glioma patients with eloquent tumours compared with standard asleep craniotomy.

The co-investigators of this study are highly renowned and very much respected in this field which gives the study tremendous strength, validity and international "robustness". This will be an important study to address the issue of whether mapping techniques (either asleep or awake) leads to better patient outcomes (including improved EOR rates, lower disability and longer survival) compared with asleep craniotomy for eloquently located high-grade gliomas. I particularly liked the authors' use of an onco-functional score that could have a lot of potential in helping predict a threshold of "maximal safe resection" with optimal survival and quality of life benefits.

With all due respect to the authors, I hope they don't mind me highlighting a few minor typographical errors and ask a few questions that I'm sure they can easily answer in terms of study design.

The study is quite similar to the SAFE trial and in the technical details in this study two arms are mentioned instead of three. Could this be amended please in the methods and results sections? Also I wondered if the statistical analysis needs to be reviewed by a statistician as the SAFE study has 2 study arms while the PROGRAM study has three.

Valuable comment. We have amended the differences between the SAFE trial and PROGRAM study. The statistician of the SAFE study has indeed performed a new power analysis for the PROGRAM study.

There are a few other minor typo errors outlined in the methods and results section. Abstract: spelling mistake "mppaing".

Change "Though" to "However, in excess of 50% of gliomas

Adjusted accordingly

Introduction: use "breadth" not "breath" Methods:

nasal "cannula" not "canular"

Adjusted accordingly

Study procedure:

Could "Description of **presenting** symptoms...." at follow-up assessments be replaced with "Description of any current (new or ongoing) symptoms...?" (use of the word "Presenting" implies symptoms at the time of diagnosis).

We have adjusted the section "study procedures".

In terms of questions about the study design I wondered about the following:

Primary outcome measures: will there be any analysis of non-enhancing tumour volume pre- and post-surgery?

Thank you for this comment. The volumetric analyses will focus on both the contrast-enhancing part of the tumor. Adjusted accordingly (see: study procedures, surgery CRF).

How useful is asleep ISM for tumours near speech centres?

Asleep mapping will not be performed in patients with speech-eloquent tumors. Adjusted accordingly (see: study design).

Will there be stratification or subgroup analysis based on tumour volume and WHO grade III/IV?

Yes, there will be multiple subgroup analyses. See: primary parameters

What if an awake craniotomy has to be converted to an asleep craniotomy? I presume this patient will then be included in the asleep group??

Yes, this patient will then be included in the asleep group.

VERSION 2 – REVIEW

REVIEWER	Ruiz-Garcia, Henry
REVIEW RETURNED	Mayo Clinic, Neurosurgery 18-Apr-2021
GENERAL COMMENTS	I congratulate the authors for the important effort they are putting together. Looking forward to seeing the results.

REVIEWER	Flannery, Thomas
REVIEW RETURNED	Royal Victoria Hospital, Neurosurgery 14-Apr-2021

GENERAL COMMENTS	Thank you for taking the suggestions on board. Happy to approve the revised manuscript for publication
---

VERSION 2 – AUTHOR RESPONSE

Thank you in advance for your swift handling.